# Clinical and cost-effectiveness of individualised (early) patient-directed rehabilitation versus standard rehabilitation after surgical repair of the rotator cuff of the shoulder: protocol for a multicentre, randomised controlled trial with integrated Quintet Recruitment Intervention (RaCeR 2)

Bruno Mazuquin [1], Maria Moffatt,[2] Alba Realpe [3,4] Rachelle Sherman,[5] Katie Ireland,[5] Zak Connan,[5] Jack Tildsley,[5] Andrea Manca [6] Vijay Singh Gc [7] Nadine E Foster,[8] Jonathan Rees,[9,10] Steven Drew,[11] Marcus Bateman [12] Apostolos Fakis,[5] Malin Farnsworth,[13] Chris Littlewood [14]

For numbered affiliations see end of article.

**Correspondence to**
Dr Bruno Mazuquin;
b.mazuquin@mmu.ac.uk

## ABSTRACT

**Introduction** Despite the high number of operations and surgical advancement, rehabilitation after rotator cuff repair has not progressed for over 20 years. The traditional cautious approach might be contributing to suboptimal outcomes. Our aim is to assess whether individualised (early) patient-directed rehabilitation results in less shoulder pain and disability at 12 weeks after surgical repair of full-thickness tears of the rotator cuff compared with current standard (delayed) rehabilitation.

**Methods and analysis** The rehabilitation after rotator cuff repair (RaCeR 2) study is a pragmatic multicentre, open-label, randomised controlled trial with internal pilot phase. It has a parallel group design with 1:1 allocation ratio, full health economic evaluation and quintet recruitment intervention. Adults awaiting arthroscopic surgical repair of a full-thickness tear are eligible to participate. On completion of surgery, 638 participants will be randomised. The intervention (individualised early patient-directed rehabilitation) includes advice to the patient to remove their sling as soon as they feel able, gradually begin using their arm as they feel able and a specific exercise programme. Sling removal and movement is progressed by the patient over time according to agreed goals and within their own pain and tolerance. The comparator (standard rehabilitation) includes advice to the patient to wear the sling for at least 4 weeks and only to remove while eating, washing, dressing or performing specific exercises. Progression is according to specific timeframes rather than as the patient feels able. The primary outcome measure is the Shoulder Pain and Disability Index total score at 12-week postrandomisation. The trial timeline is 56 months in total, from September 2022.

**Trial registration number** ISRCTN11499185.

## STRENGTHS AND LIMITATIONS OF THIS STUDY

⇒ RaCeR 2 is a large randomised controlled trial investigating the clinical and cost-effectiveness of individualised (early) patient-directed rehabilitation after surgery to repair the torn rotator cuff of the shoulder.
⇒ We will explore and address barriers to recruitment with the quintet recruitment intervention to optimise recruitment.
⇒ In addition to self-reported outcome measures, participants will have an ultrasound scan at 12 months to assess rotator cuff repair integrity.
⇒ The parallel health economic analysis will assess the cost-effectiveness of individualised (early) patient-directed rehabilitation in comparison to standard rehabilitation.

## INTRODUCTION

Shoulder pain associated with a rotator cuff tear can significantly affect a person's quality of life.[1] The number of operations to repair rotator cuff tears has increased over time.[2] In 2018/2019, direct treatment costs in the UK National Health Service (NHS) amounted to £56.7 million.[3] Following surgery, rehabilitation is provided to support patients' recovery. Current standard rehabilitation in the UK NHS typically includes sling immobilisation for approximately 1 month. This has not changed for over 20 years and may be contributing to suboptimal outcomes.[4]

Our systematic review of 20 randomised controlled trials (RCTs) compared the effectiveness of early versus standard postoperative rehabilitation. We found no difference between the approaches for shoulder pain and disability up to 12 months, but early rehabilitation significantly improved range of movement.[5] Rotator cuff re-tear after surgery is a concern for clinicians and underpins the rationale for more cautious approaches to postoperative rehabilitation. We found no difference in repair integrity between rehabilitation approaches, but rehabilitation protocols varied and approaches described as early mobilisation were more reflective of standard rehabilitation in the UK.[5]

In our RaCeR pilot, 73 patients from 5 NHS hospitals were randomised to individualised (early) patient-directed rehabilitation (EPDR) (advice to remove the shoulder sling as soon as able and move as symptoms allow) or standard rehabilitation (sling immobilisation for 4 weeks). Participants in the EPDR reported less shoulder pain and disability, returned to driving 18 days faster, had 4 fewer days lost from work over 12 weeks and fewer re-tears (30% vs 41%).[6] These findings from our RaCeR pilot, combined with our favourable assessment of feasibility and an evaluation of the need for evidence using principles of value of information to research prioritisation, provided the basis for the fully powered RCT (RaCeR 2).

## Objectives

Our hypothesis is that individualised EPDR is superior to standard rehabilitation for shoulder pain and disability, measured using the Shoulder Pain and Disability Index (SPADI)[7] at 12 weeks postrandomisation. The aim of RaCeR 2 is to assess the clinical and cost-effectiveness of individualised EPDR compared with NHS standard rehabilitation for pain and disability at 12 weeks after rotator cuff repair. The objectives include:

► Understanding and mitigating barriers to recruitment.
► Shoulder pain and disability at 6 and 12 months, quality of life, time to return to drive and usual activities including work, further healthcare use, repair integrity and adverse events (AEs) to 12 months.
► Within-trial cost-consequence analysis from an NHS and personal social services perspective and model-based long-term cost-effectiveness analysis.

## Trial design

Pragmatic multicentre, open-label, RCT with internal pilot. It follows a parallel group design with 1:1 allocation ratio, with full economic evaluation and integrated quintet recruitment intervention (QRI).[8]

## METHODS: PARTICIPANTS, INTERVENTIONS AND OUTCOMES

This protocol paper follows the Standard Protocol Items: Recommendations for Interventional Trials.[9]

## Study setting

A minimum of 24 NHS orthopaedic and physiotherapy services across the UK will be opened for recruitment. The internal pilot will last 6 months (June–November 2023).

**Table 1** Internal pilot progression criteria

| Progression criteria | Red (stop)* <66% | Amber (amend)† ≥66%–99% | Green (go)‡ 100% |
|---|---|---|---|
| Average recruitment rate/site/month | <0.7 | 0.7–1.0 | 1.1 |
| Sites open | <12 | 12–17 | 18 |
| Participants recruited | <50 | 50–96 | 97 |

*Red: halt, do not progress to main study.
†Amber: review areas of weakness and make amendments accordingly.
‡Green: no action required, continue to main study.

The trial monitoring group (TMG) in consultation with the trial steering committee (TSC) will assess study progress and decide on progression based on the criteria in table 1.

## Eligibility criteria
### Inclusion criteria

► Adults (18 years or older) awaiting arthroscopic surgical repair of a full-thickness tear of their shoulder rotator cuff, of any size.
► Able to return to the recruiting centre or affiliated site for rehabilitation supported by physiotherapists trained to deliver the study interventions.

### Exclusion criteria

► Do not have a full-thickness tear at surgery and/or arthroscopic repair is not undertaken.
► Unable to provide informed consent.
► Taking part in another research study that mandates a specific postoperative rehabilitation pathway.

## Recruitment and informed consent

Patients listed for rotator cuff repair surgery are screened and assessed for eligibility by trained local hospital site staff. Once an eligible patient has been identified and has been allocated a date for surgery, they are provided with an information pack about the study (including the optional QRI; more details below) and consent forms. Patients will be given the opportunity to discuss RaCeR 2 with support from an interpreter as required. We provide translated information sheets in Arabic, Bengali, Korean, Nepali, Polish, Punjabi, Turkish, Urdu and Welsh; potential recruiting sites identified these languages as the most common languages spoken other than English in their areas. Recruiters will follow up with the patient to discuss the study and answer any questions. Patients may consent to participate in neither, either or both the QRI and RaCeR 2 trial. Separate QRI clauses relating to recording of discussions about the study are included within the consent form. The process of gaining informed consent may be wholly or partly undertaken remotely or in-person depending on local site and patient circumstances. If it is not possible to get written consent, for example, if the patient is not returning to the recruiting site prior to surgery, verbal remote consent will be acceptable to avoid unnecessary burden for the patients and site staff. The consent form is

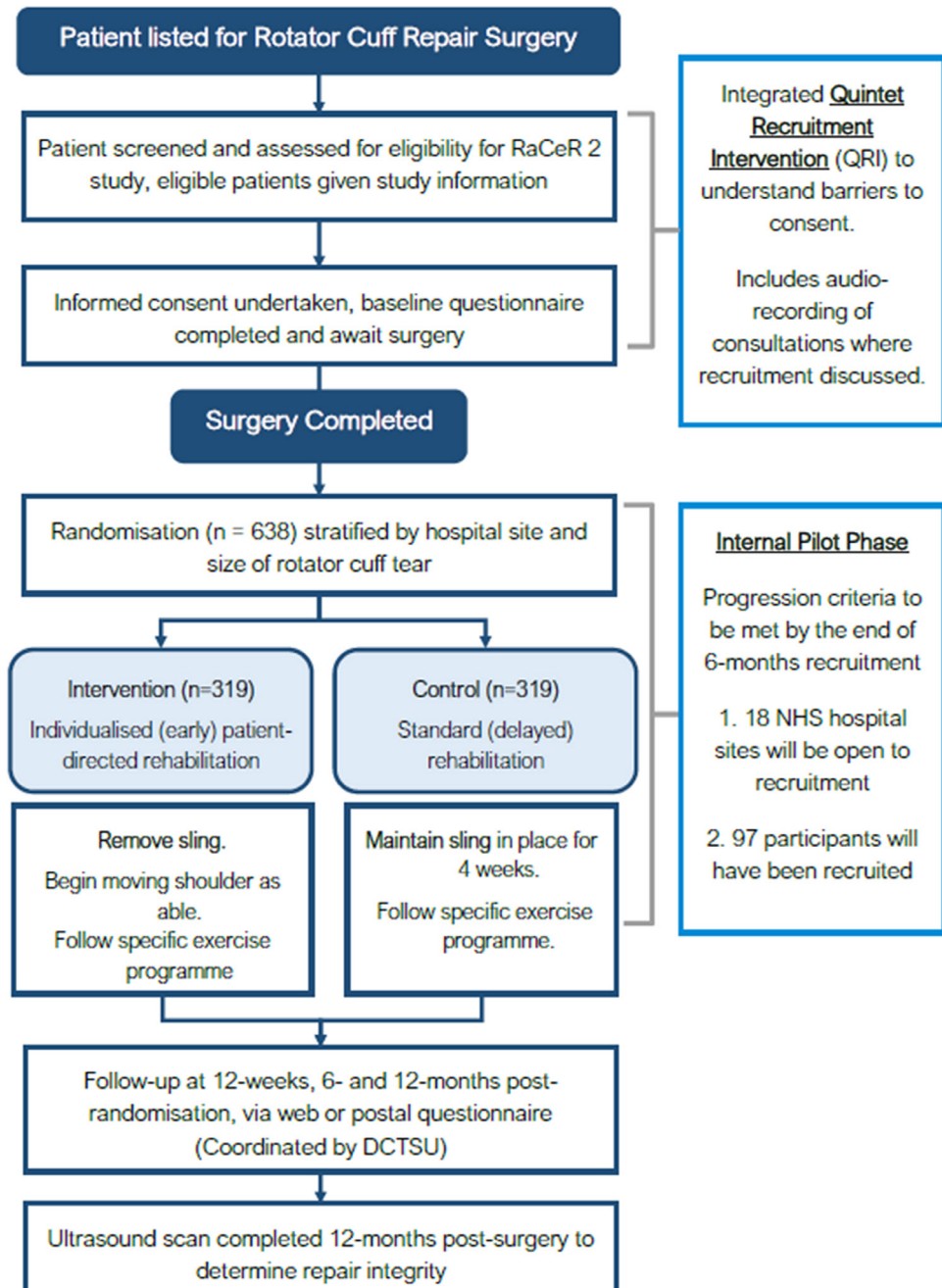

**Figure 1** Study flow diagram. DCTSU, Derby Clinical Trials Support Unit; NHS, National Health Service.

completed by the recruiter indicating that consent was taken verbally and a copy provided to the participant. This is the same for patients who consent to the audio recording of their discussion (QRI) but not to participating in RaCeR 2. Consent is fully documented within the patient's medical notes, including the method of consent (remote/in-person and written/verbal). Figure 1 shows the study flow diagram.

### Interventions
#### Individualised EPDR
EPDR is an individualised approach where shoulder movement, sling removal and exercise are progressed as the participant feels able within the context of their own pain experience and tolerance. Individualised EPDR includes

advice to the patient from a physiotherapist within 24 hours following surgery to remove their sling and gradually begin to actively use their arm as they feel able and within acceptable limits of pain. The advice to remove the sling is complemented by an exercise programme supervised by a physiotherapist and practised at home. After the first session with the physiotherapist, participants access follow-up with a physiotherapist according to usual care agreements. Follow-up sessions can be either face to face or remote.

#### Standard (delayed) rehabilitation
Standard (delayed) rehabilitation includes advice to the patient from a physiotherapist within 24 hours following surgery to wear their sling for 4 weeks except for when

eating, washing, dressing or undertaking the exercises prescribed. After the first session with the physiotherapist, participants access follow-up with a physiotherapist. The exercise programmes will be staged as follows[6]:

Stage 1: Fully assisted (passive) shoulder movement.

Stage 2: Partially assisted (active assisted) with progression to full non-assisted (active) shoulder movement.

Stage 3: Resisted static exercises (isometric).

Stage 4: Resisted exercises through shoulder range of movement (dynamic) within limits of pain progressing to functional restoration.

### Difference between current and planned care pathways

Participants in both groups agree to the number of rehabilitation sessions with their physiotherapists; there is no prespecified number of sessions. It is expected that approximately five follow-up appointments will be scheduled over the 12-week period following surgery. This means that both treatments are delivered within the parameters of current NHS physiotherapy provision. The key difference between the two rehabilitation approaches is that the individualised EPDR promotes an approach to rehabilitation which reflects patient factors including pain, preoperative levels of function and psychological well-being. It aims to promote self-efficacy whereby the patients feel they have increased control over their recovery. Both groups will start with stage one of the specific exercise programme but the intervention group will be supported to progress through the stages as they feel able. The control group will remain at stage 1 for a minimum of 4 weeks. Patients undergoing individualised EPDR are invited to resume activities in line with their individual progress rather than preset timescales. Patients receiving standard rehabilitation will progress through stages based on specific time frames after surgery; stage 1 (0–4 weeks), stage 2 (4–6 weeks), stage 3 (6–8 weeks) and stage 4 (8–12 weeks). Surgeons and physiotherapists will treat patients in both arms of the trial and multiple clinicians will be involved in patients' treatment in each arm.

### Criteria for discontinuing or modifying allocated interventions

There are no specific criteria to discontinue or modify the allocated interventions. Participants can withdraw at any time. If they opt for withdrawing from the allocated treatment, they will receive standard NHS care.

### Strategies to improve adherence to interventions

Participants are supported by a physiotherapist to remove their sling as they feel able or to maintain the sling in place for 4 weeks, depending on their allocated intervention. Participants are also supported by a physiotherapist to adhere to their prescribed exercise programme through the individual consultations and a study specific manual and website that detail the exercises and progressions. Participants are also asked to complete a sling use diary to record their time out of the sling at regular periods throughout the day for 4 weeks postrandomisation.

### Relevant concomitant care permitted or prohibited during the trial

No concomitant care is prohibited in RaCeR 2. Other healthcare use will be collected during the trial, summarised and described.

### Provisions for post-trial care

None beyond standard NHS care.

### Outcomes

Figure 2 presents the trial schedule of outcomes and assessments. On receipt of informed consent (online supplemental material 1), questionnaires are completed at baseline (before surgery), and at 12 weeks, 6 and 12 months after randomisation. At baseline, the questionnaire will include demographic data (eg, date of birth, sex and ethnicity), the SPADI and EuroQol five dimensions five levels (EQ-5D-5L).

| | | | | Timepoint | | | |
|---|---|---|---|---|---|---|---|
| | Screening | Baseline | Surgery | 4 weeks | 12 weeks | 6 months | 12 months |
| **ENROLMENT** | | | | | | | |
| Eligibility | X | | | | | | |
| Participant invitation | X | | | | | | |
| Screening data collected | X | | | | | | |
| Recording of recruitment appointment/ Informed consent (QRI) | | X | | | | | |
| Randomisation | | | X | | | | |
| **INTERVENTIONS** | | | | | | | |
| Individualised (early) patient-directed rehabilitation | | | | •———————————• | | | |
| Standard rehabilitation | | | | •———————————• | | | |
| **ASSESSMENTS** | | | | | | | |
| Baseline questionnaire | | X | | | | | |
| SPADI | | X | | | X | X | X |
| EQ-5D-5L | | X | | | X | X | X |
| Sling-use diary | | | | •———• | | | |
| Adverse event questionnaire | | | | | X | X | X |
| Adverse event assessments (by clinicians) | | | | •————————————————• | | | |
| Healthcare resources use | | | | | X | X | X |
| Ultrasound imaging | | | | | | | X |
| Assessment of treatment fidelity (by PIs) | | | | | | | X |

**Figure 2** Trial schedule of assessments and outcomes. EQ-5D-5L, EuroQol five dimensions five levels; PI, principal investigator, QRI, quintet recruitment intervention, SPADI: Shoulder Pain and Disability Index.

## Primary outcome measure

Shoulder pain and disability at 12 weeks postrandomisation will be measured using the SPADI. The SPADI is a validated self-report measure,[7] it was more sensitive and responsive than the Oxford Shoulder Score in our RaCeR pilot and is the most used outcome measure in RCTs evaluating interventions for shoulder disorders.[6]

## Secondary outcome measures

► Shoulder pain and disability at 6 and 12 months postrandomisation will be measured using the total SPADI score.
► Health-related quality of life at 12 weeks, 6 and 12 months postrandomisation will be measured using the EQ-5D-5L.
► Time to return to usual activities, including work and driving, will be measured via self-report questionnaire at 12 weeks, 6 and 12 months.
► Healthcare resource use at 12 weeks, 6 and 12 months will be measured via self-report questionnaire.
► Rotator cuff repair integrity (evidence of full-thickness re-tear; yes/no) at 12 months will be assessed via diagnostic ultrasound scan.
► Number and nature of AEs at 12 weeks, 6 and 12 months will be measured via self-report questionnaire and clinician report.
► Self-report time out of the sling, measured in hours, over 4 weeks postsurgery via self-report diary.

## Participants timeline

See figures 1 and 2.

## Sample size

The sample size calculation was based on total SPADI score at 12 weeks, powered to detect a minimal clinically important difference of 8 points between groups.[10] Assuming an SD of 30 (the upper 80% confidence limit from our pilot study)[6] at 90% power and significance level 5%, and using an independent t-test, results in 297 participants being needed per group (594 in total). However, using Analysis of Covariance (ANCOVA) (primary analysis), adjusting for the baseline SPADI score, where correlation (r) between baseline and 12 weeks=0.2 (data from pilot RaCeR RCT), the sample size was adjusted by (1-r2) plus one extra participant per group to 574 in total.[11] In addition, adjusting for 10% non-response of SPADI questionnaire at 12 weeks, a target of 319 participants should be randomised per group, 638 in total.

## Recruitment strategies

### The quintet recruitment intervention

We will implement the QRI aiming to optimise recruitment.[8] Although our RaCeR pilot recruited 39% of those eligible, we anticipate challenges to recruitment in the main trial due to: (1) hesitance by surgeons to randomise patients (particularly older patients with larger rotator cuff tears) and (2) challenges in participants accepting the randomised allocation due to perceived risks of individualised EPDR.

The QRI has been applied to over 25 RCTs to date, leading to insights about individual and generic recruitment issues and the development of targeted strategies to improve recruitment rates.[12 13] Rather than simply increasing the numbers of patients recruited, the QRI will aim to reduce 'missed opportunities' for enrolling eligible patients, while safeguarding informed consent. We will draw on insights from previous application of QRI methods in RCTs, and the latest recruitment-related evidence to develop materials and pre-emptive training which will support participant recruitment from the outset of RaCeR 2. Once sites open to recruitment, we will proceed to implement the QRI in two phases:

Phase 1: We will investigate recruitment issues that transpire 'in real time' throughout the remainder of the scheduled recruitment period. We will use mixed methods to investigate actual (rather than anticipated) issues hindering recruitment as the trial proceeds. Data collection will include:
► Semistructured interviews with individuals involved in recruitment ('recruiters').
► Audiorecorded discussions between recruiters and potential participants about RaCeR 2.
► Mapping of recruitment pathways and screening log analysis.

Findings from these sources will be triangulated to generate an in-depth understanding of the 'root-causes' of key recruitment issues.

Phase 2: Using the results from phase 1, the QRI team will work closely with the TMG and patient and public involvement (PPI) group to design and implement 'actions' to optimise recruitment. Actions may be applicable to all sites, specific sites or individual recruiters and will aim to increase the number of eligible patients approached, and/or improve conversion rates while safeguarding informed consent. The QRI phases will run iteratively. New avenues of enquiry will emerge throughout the conduct of the QRI, through discussion in feedback meetings and continued monitoring of screening logs.

We will pay close attention to screening log data before/after QRI actions to formatively evaluate the impact of actions, and the need for further investigation (phase 1) or actions (phase 2). Part of the QRI will entail up-front training for site staff as they open to recruitment. This training will evolve to become increasingly focused as we develop our understanding of recruitment issues, with a view to ensuring sites that open in the latter stages of the trial benefit from the QRI insights that have emerged to date.

## Assignment of interventions: allocation

### Sequence generation

On completion of surgery, participants are randomised using minimisation. Participants are allocated on a 1:1 ratio, stratified by recruiting site and rotator cuff tear size; small (<1 cm), medium (1–3 cm), large/massive (>3 cm) or unknown.

### Concealment mechanism

To ensure allocation concealment, randomisation is coordinated by Derby Clinical Trials Support Unit (DCTSU) remotely via an online randomisation system.

### Implementation

The allocation sequence is generated by an online randomisation system. Following surgery, the local site team will explain to the participant their randomised allocation as well as other routine post-operative requirements. An exercise manual is provided to all participants, along with the sling diary. Participants will complete the diary with the amount of time (hours and minutes) they were not wearing the sling at regular periods throughout the day.

### Assignment of interventions: blinding (masking)

RaCeR 2 is an open-label RCT. No blinding of participants, clinicians, research team or oversight committees is in place.

### Data collection, management and analysis
### Plans for assessment and collection of outcomes

Following consent, the baseline questionnaire will be completed prior to surgery in-person or remotely. Completion of the baseline questionnaire will require input from local site staff and participants. The questionnaire will include the SPADI and EQ-5D-5L validated questionnaires and demographic data. The SPADI has 13 items divided into 2 subscales: pain (5 items) and disability (8 items). The responses are indicated on a Visual Analogue Scale (0=no pain/no difficulty and 10=worst imaginable pain/so difficult it requires help). The items are summed and converted to a total score out of 100, a high score indicates greater pain and disability.[7] The EQ-5D-5L is a generic measure of health-related quality of life. It provides a single index value for health status that can be used in a clinical or health economic evaluation.[14] The EQ-5D-5L consists of questions relating to five health domains (mobility, self-care, usual activities, pain/discomfort and anxiety/depression) and respondents rate their degree of impairment using five response levels (no problems, slight, moderate, severe or extreme problems). The EQ-5D-5L is the National Institute for Health and Care Excellence's preferred measure of health-related quality of life in adults.

Follow-up questionnaires, including SPADI, EQ-5D-5L, self-report questionnaire for healthcare resource use, time to return to usual activities (including work) and any AEs, will be completed at 12 weeks, 6 and 12 months after randomisation (+4 weeks visit window to allow for reminders). This process will be coordinated centrally by the DCTSU. Follow-up questionnaires will be available in paper or electronic format. At 12 months following surgery, participants will be asked to undergo an ultrasound scan.

### Plans to promote participant retention and complete follow-up

If participants do not complete their questionnaires at the expected timepoints, they will be contacted at 2 weeks and a minimum data collection (SPADI and AEs) will be attempted via telephone at 3 weeks.

### Data management

A secure electronic software platform (Dacima) will be used to store participant study data. Each participant is assigned a participant ID for use on study forms, other study documents and the electronic database.

### Confidentiality

All documents will be stored safely in confidential conditions in accordance with the Data Protection Act 2018 and UK General Data Protection Regulation and retained according to national legislation.

### Statistical methods
### Primary and secondary outcome analysis

Primary analyses will be conducted according to the intention-to-treat analysis group. ANCOVA will be used to compare total SPADI scores between individualised EPDR versus standard rehabilitation at 12 weeks after randomisation, adjusting for baseline SPADI score.

Among other secondary analyses, time to return to usual activities (work and driving) will be analysed using Kaplan-Maier curves and log rank test. Logistic regression will be undertaken to test the association between treatment groups and re-tear at 12 months. Linear regression will be used to test the association between treatment groups and time out of the sling over 4 weeks. Repeated measures ANCOVA will be used to test if any treatment effect exists and has been maintained up to 12 months in terms of SPADI and EQ-5D-5L scores. ANCOVA will be used to compare total SPADI and EQ-5D-5L scores between the treatment groups at 6 and 12 months adjusting for baseline scores. Safety analysis will be undertaken based on the per-protocol analysis group. The presence of AEs/serious AEs and problems after surgery will be compared between the two groups at 12 weeks, 6 and 12 months using $\chi^2$ test.

### Interim analyses

Interim descriptive analysis will be undertaken at 6 months from the start of recruitment to assess the progression criteria of the internal pilot phase. This will not include any comparison of the patient reported outcomes between the randomised groups.

### Methods for additional analyses

Exploratory subgroup analysis will be undertaken for the primary endpoint at 12 weeks including an interaction term in the ANCOVA model of 'rotator cuff tear size' by 'treatment group'.

### Definition of analysis population relating to protocol non-adherence and any statistical method to handle missing data

Per-protocol analysis will consider patients with time out of the sling of 222.6 hours or more over 4 weeks compared with those with time out of the sling less than 222.6 hours

base on the cut-off values from the RaCeR pilot.[6] Missing values in the diary will not be included in the analysis.

Complete-cases analysis will be undertaken as part of the primary endpoint analyses, where cases with missing values or those completed outside the 4 weeks window will be excluded in each analysis. If substantial missing data (>10% and <20%) are observed in SPADI at 12 weeks or a key prognostic covariate for the primary analysis, then multiple imputation using chained equations will be applied. Complete-cases analysis will be undertaken for the secondary study outcomes.

### Economic analysis

The perspective for both within-trial and model-based economic analyses will be that of the NHS and personal social services.[15] The economic analysis has three phases:

1. Development of a conceptual cost-effectiveness model structure: an initial conceptual cost-effectiveness model structure will be developed to estimate the long-term costs and quality-adjusted life-year of EPDR and standard rehabilitation.
2. Within-trial cost-consequences analysis: health benefits will be quantified for changes in health-related quality of life, measured by the EQ-5D-5L. Healthcare resource use and costs observed during the trial period will be reported for each treatment group. Outcomes measured during the 12-month study period will be left undiscounted.
3. Model-based economic analysis: The long-term costs and health outcomes of EPDR and standard rehabilitation will be modelled for their impact on clinically relevant events (eg, re-tear, reoperation), updating the state-transition model developed using the RaCeR pilot with parameters derived from data collected in RaCeR 2 and (where relevant) the published literature. Long-term predicted outcomes will be discounted at 3.5% per annum.[15] The health economic analysis plan will be developed and finalised before analysis commenced and is anticipated to be disseminated in a separate publication.

### Plans to give access to the full protocol, participant-level data and statistical code

The full protocol is available at https://www.fundingawards.nihr.ac.uk/award/NIHR133874. In the first instance, further requests for data can be made via the chief investigator (CL).

### Oversight and monitoring
### Composition of the coordinating centre and TSC

The chief investigator (CL) is responsible for the conduct of the trial and will be supported by the TMG. The TMG oversees all day-to-day aspects of trial management and delivery. The independent TSC monitors the trial progress and ensures that is it is being conducted according to the protocol and the applicable regulations. The TSC has an independent chair (statistician), and four other independent members including a health economist,

physiotherapist, surgeon and two PPI representatives as well as the chief investigator (non-independent). The TSC will meet annually. The chief investigator, associate investigator, statistician and trial manager will attend the TSC meetings and report on trial progress.

### Composition of the data monitoring committee, its role and reporting structure

Given the nature of RaCeR 2, a separate data monitoring committee will not be convened and the TSC will take on the data monitoring role, as agreed by the funder.

### AE reporting and harms

Number and nature of AEs at 12 weeks, 6 and 12 months will be measured via self-report questionnaire and clinician report. AEs might include an increase in shoulder pain requiring additional care, for example, prescribed medication or injection; infection up to 12 weeks post-surgery; other shoulder disorders, for example, stiffness; rotator cuff re-rupture requiring additional care, for example, injection, physiotherapy or surgery.

### Frequency and plans for auditing trial conduct

Audits will be conducted by the sponsor (University Hospitals of Derby and Burton NHS Foundation Trust) according to their audit plan; these may be central or site audits and may be trial or process-level audits.

### Plans for communicating important protocol amendments to relevant parties

Substantial amendments will be submitted by the sponsor to relevant regulatory bodies (Research Ethics Committee and Health Research Authority) for review and approval. The amendments will only be implemented after approval and a favourable opinion has been obtained. Non-substantial amendments will be submitted to the Health Research Authority for their approval/acknowledgement.

### ETHICS AND DISSEMINATION

We were granted ethical approval by London-Stanmore Research Ethics Committee (23/LO/0195). We will disseminate findings from RaCeR 2 to stakeholders via peer-reviewed publications and presentations at national and international conferences. Our website ( www.racer2study.co.uk) will serve as a hub to videos describing the trial results to support patient and clinical decision-making.

### Patient and public involvement

PPI was embedded throughout our RaCeR pilot.[6] Our PPI group informed the choice of primary outcome, directed the timing of the intervention, the reporting of ultrasound scans and the follow-up data collection methods. They will continue to be actively involved in all stages of RaCeR 2, including development of patient-facing documents and the qualitative interview schedule for the QRI. We will work collaboratively to cocreate dissemination

materials such as blogs and social media posts accessible to members of the public. The coauthor MF is a TMG member. Our PPI group holds regular meetings, facilitated by our PPI lead (MM).

## DISCUSSION

RaCeR 2 will be the largest RCT in the world investigating rehabilitation after rotator cuff repair.[5] The findings will inform national and international clinical practice. Our primary outcome assesses pain and disability. Our comprehensive dataset will assess other outcomes of interest to the clinical community, including rotator cuff repair integrity, and the comparative cost-effectiveness of individualised EPDR versus standard rehabilitation.

## Study status

The RaCeR 2 trial (protocol version 2.2, 14 April 2023) opened to recruitment on 1 June 2023 and is scheduled to remain open until 31 May 2025.

**Author affiliations**
[1]Health Professions, Manchester Metropolitan University, Manchester, UK
[2]School of Allied Health Professios and Nursing, University of Liverpool, Liverpool, UK
[3]Population Health Sciences, Bristol Medical School, University of Bristol, Bristol, UK
[4]NIHR Bristol Biomedical Research Centre, University Hospitals Bristol and Weston NHS Foundation Trust and University of Bristol, Bristol, UK
[5]Derby Clinical Trials Support Unit, Royal Derby Hospital, Derby, UK
[6]Centre for Health Economics, York University, York, UK
[7]School of Human and Health Sciences, University of Huddersfield, Huddersfield, UK
[8]STARS Education and Research Alliance, Surgical Treatment and Rehabilitation Service (STARS), The University of Queensland and Metro North Health, Saint Lucia, Queensland, Australia
[9]Nuffield Department of Orthopaedics Rheumatology and Musculoskeletal Sciences, University of Oxford, Oxford, UK
[10]NIHR Oxford Biomedical Research Centre, Oxford, UK
[11]University Hospitals Coventry & Warwickshire NHS Trust, Coventry, UK
[12]Derby Shoulder Unit, University Hospitals of Derby and Burton NHS Foundation Trust, Derby, UK
[13]Patient and Public Involvement Representative, England, UK
[14]Allied Health, Social Work & Wellbeing, Faculty of Health Social Care and Medicine, Edge Hill University, Ormskirk, UK

**Contributors** CL, MB, NEF, BM, MM, AR, AM, VSG, JLR, SD and AF conceived of the study and were involved, alongside RS, KI and MF in developing the design and protocol. CL, BM, MM, AM, VSG, AR, NEF, JLR, SD, MB and AF secured funding for the study. BM drafted the manuscript and all other authors reviewed and provided feedback on drafts. All authors read and approved the final version of the manuscript.

**Funding** This study is funded by the NIHR Health Technology Assessment programme (Ref:133874). The views expressed are those of the author(s) and not necessarily those of the NIHR, the Department of Health and Social Care or the NHS.

**Disclaimer** The views expressed are those of the author(s) and not necessarily those of the NIHR, the Department of Health and Social Care, or the NHS.

**Competing interests** All authors declare support from the National Institute of Health and Care Research for the present manuscript. AM is non-executive director of the ISPOR. JLR is Past President of the British Elbow and Shoulder Society (2021-2023). SD reports education consultancy contracts with Stryker, Smith and Nephew and Arthrex for teaching and training. SD is the president of the British Elbow and Shoulder Society (2023–2025).

**Patient and public involvement** Patients and/or the public were involved in the design, or conduct, or reporting, or dissemination plans of this research. Refer to the Methods section for further details.

**Patient consent for publication** Not applicable.

**Provenance and peer review** Not commissioned; externally peer reviewed.

**ORCID iDs**
Bruno Mazuquin http://orcid.org/0000-0003-1566-9551
Alba Realpe http://orcid.org/0000-0001-9502-3907
Andrea Manca http://orcid.org/0000-0001-8342-8421
Vijay Singh Gc http://orcid.org/0000-0003-0365-2605
Marcus Bateman http://orcid.org/0000-0002-3203-506X
Chris Littlewood http://orcid.org/0000-0002-7703-727X

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
