## [Reviewer comments · BMJ Open]

ARTICLE DETAILS

TITLE (PROVISIONAL)	Clinical and cost-effectiveness of individualised (early) patient-directed rehabilitation versus standard rehabilitation after surgical repair of the rotator cuff of the shoulder: protocol for a multi-centre, randomised controlled trial with integrated Quintet Recruitment Intervention (RaCeR 2).
AUTHORS	Mazuquin, Bruno; Moffatt, Maria; Realpe, Alba; Sherman, Rachelle; Ireland, Katie; Connan, Zak; Tildsley, Jack; Manca, Andrea; Gc, Vijay; Foster, Nadine; Rees, J L; Drew, Steven; Bateman, Marcus; Fakis, Apostolos; Farnsworth, Malin; Littlewood, Chris

VERSION 1 – REVIEW

REVIEWER	Maestroni, Luca Smuoviti
REVIEW RETURNED	10-Nov-2023

GENERAL COMMENTS	Thank you for asking me to review this study protocol. The authors have clearly described their methodology, and they have far more expertise than myself in conducting randomized controlled trials, which is why I cannot raise any concern regarding their selected methodological approach. However, the rehabilitation program is not described sufficiently in my opinion. They stated "individualized" program, which seems to be based on patients' pain experience and tolerance until they resume their activities. I do not see any indication in terms of exercise frequency, dosage and prescription to recover, for example, muscle volume and strength following surgical repair. Are these not "individualized" strategies for better care? I do understand that there will be a diary, but it is unclear which rationale based on exercise physiology (not only pain experience and tolerance) will be adopted by the treating clinicians. Also, there is no understanding on how shoulder specific objective measurements (e.g. TROM, ER, IR strength) will be used to guide the "individual" / "specific" program? What are the aimed discharge criteria and/or rehabilitation goals? Again, are these not as important following surgery as pain tolerance and pain experience? If such approach would be used following Achilles Tendon Repair/Reconstruction, is resuming normal activities using pain tolerance and pain experience a valid indication of "individualized" care that accounts for a thorough BPS approach? I do believe that rehabilitation specialists should be more accurate in describing such important parameters currently. I don't think that the program and/or the level of knowledge/expertise of the treating physiotherapists is well specified in this protocol. I do understand the main aim is to assess difference in early versus delayed
--

	rehabilitation, but I believe that the term "individualized" rehabilitation should either be removed or better explained. Similarly the use of the term "specific" exercise program does not seem to have any strong underpinning rationale. Overall, I commend the authors for such important work and I hope my comments may be helpful to improve the quality of this protocol.
--	---

REVIEWER	Smythe, A Monash University, Physiotherapy
REVIEW RETURNED	18-Nov-2023

GENERAL COMMENTS	Overall this is exceptionally written and thought out. I just had several small queries. 1.) I think you need to define early vs standard treatment earlier in the article and when is it discussed outline exactly what the standard protocol is currently as it took a long time to get there reading through the methods. Page 3 Line 56 – fewer re-tears. What timeline was this measured on? Was this at 12 months? Page 3 Line 51-52 – from the previous study was the average time in sling calculated for both groups? Or the average time to progress to the next stage? Lastly, the way it is setup. If both groups get 5 sessions with physiotherapy, but the early group would theoretically advance through the stages quicker, does this mean that in the last 1-2 sessions they may be given more advanced stage 4 exercises? As in will they almost get a stage 5 set of exercises with physiotherapists progressing those in stage 4 that the standard group may not get? Or is the exercise protocol the same for every patient and there is no extra progression if a patient advances quickly through the stages?
--

VERSION 1 – AUTHOR RESPONSE

Reviewer: 1

Dr. Luca Maestroni, Smuoviti

Comments to the Author:

C: Thank you for asking me to review this study protocol. The authors have clearly described their methodology, and they have far more expertise than myself in conducting randomized controlled trials, which is why I cannot raise any concern regarding their selected methodological approach.

A: Thank you for your comment.

C: However, the rehabilitation program is not described sufficiently in my opinion. They stated "individualized" program, which seems to be based on patients' pain experience and tolerance until they resume their activities. I do not see any indication in terms of exercise frequency, dosage and prescription to recover, for example, muscle volume and strength following surgical repair. Are these not "individualized" strategies for better care? I do understand that there will be a diary, but it is unclear which rationale based on exercise physiology (not only pain experience and tolerance) will be adopted by the treating clinicians.

A: Thank you for your comment. Please be aware that this is a protocol reporting a trial that has been funded. Hence, we are mandated to evaluate the intervention as described and approved by the funder. At this stage there is very limited scope to make changes. We are funded to test the clinical and cost-effectiveness of individualised early patient directed rehabilitation as described in the protocol. Recovery of muscle volume and strength is not the key focus of this approach. As we describe in the protocol, sling removal and movement progression is directed by the patient as they feel able. At this stage, we have opted not to provide the detailed treatment manuals given to patients and therapists, which includes some guidance on exercise dosage. This is a purposeful decision given the stage of this research. We will provide this detail when reporting the results of the trial in due course.

We are also limited by word count and, as such, feel the description is appropriate for a published research protocol.

C: Also, there is no understanding on how shoulder specific objective measurements (e.g. TROM, ER, IR strength) will be used to guide the "individual" / "specific" program?

A: Yes, that is correct. We are not incorporating objective measurements. Our clinical outcomes are patient reported in nature and as described in the previous point, patients progress through their rehabilitation as they feel able.

C: What are the aimed discharge criteria and/or rehabilitation goals? Again, are these not as important following surgery as pain tolerance and pain experience? If such approach would be used following Achilles Tendon Repair/Reconstruction, is resuming normal activities using pain tolerance and pain experience a valid indication of "individualized" care that accounts for a thorough BPS approach?

A: Discharge criteria and rehabilitation goals are individually negotiated between patient and therapist and are variable. Again, this aligns with the patient-directed nature of the intervention.

C: I do believe that rehabilitation specialists should be more accurate in describing such important parameters currently. I don't think that the program and/or the level of knowledge/expertise of the treating physiotherapists is well specified in this protocol. I do understand the main aim is to assess difference in early versus delayed rehabilitation, but I believe that the term "individualized" rehabilitation should either be removed or better explained.

A: We agree, please see response to previous comments including limitations imposed by word count.

C: Similarly the use of the term "specific" exercise program does not seem to have any strong underpinning rationale.

A: As described in the protocol, patients do undertake a specific exercise programme rather than a general exercise programme. So, the term specific seems appropriate and we have maintained that in the manuscript.

C: Overall, I commend the authors for such important work and I hope my comments may be helpful to improve the quality of this protocol.

A: Thank you for your comments.

Reviewer: 2

Dr. A Smythe, Monash University, Lively Physiotherapy

Comments to the Author:

C: Overall this is exceptionally written and thought out. I just had several small queries.

A: Thank you for your comment.

C: 1.) I think you need to define early vs standard treatment earlier in the article and when is it discussed outline exactly what the standard protocol is currently as it took a long time to get there reading through the methods.

A: The protocol is structured according to the SPIRIT checklist.

C: Page 3 Line 56 – fewer re-tears. What timeline was this measured on? Was this at 12 months?

A: In our RaCeR pilot, re-tear was assessed at 3 months. This is reported in the open access paper that we reference in the protocol.

C: Page 3 Line 51-52 – from the previous study was the average time in sling calculated for both groups? Or the average time to progress to the next stage?

A: Details of sling use in the RaCeR pilot are reported in the open access paper that we reference in the protocol.

C: Lastly, the way it is setup. If both groups get 5 sessions with physiotherapy, but the early group would theoretically advance through the stages quicker, does this mean that in the last 1-2 sessions they may be given more advanced stage 4 exercises? As in will they almost get a stage 5 set of exercises with physiotherapists progressing those in stage 4 that the standard group may not get? Or is the exercise protocol the same for every patient and there is no extra progression if a patient advances quickly through the stages?

A: As described in the protocol, progression is individualised to enable patients to achieve their individual goals. Numbers of treatment sessions and focus are negotiated between patient and therapist. We are not mandating an end-point for the rehabilitation in this research

VERSION 2 – REVIEW

REVIEWER	Maestroni, Luca Smuoviti
REVIEW RETURNED	06-Feb-2024
GENERAL COMMENTS	Good work